# Effect of Local Annealing on Magnetic Flux Distribution and Noise in a Micro-Generator with Amorphous Shell

**DOI:** 10.3390/mi13122086

**Published:** 2022-11-26

**Authors:** Chang-Hung Hsu

**Affiliations:** Department of Mechanical Engineering, Asia Eastern University of Science and Technology, No. 58, Section 2, Sichuan Road, Banqiao District, New Taipei 220, Taiwan; chshiu@mail.aeust.edu.tw; Tel.: +886-2-77388000 (ext. 3133); Fax: +886-2-77380145

**Keywords:** generator, amorphous alloy, housing, magnetic flux density, noise

## Abstract

This study examined micro-alternators with two different housing structures––an uncoated shell and a shell coated with an iron-based amorphous-alloy soft magnetic material. The electromagnetic power and noise characteristics of generators with these shell structures were measured and analyzed. The material used for the shell coating was the SA1 amorphous alloy. The magnetic property of the SA1 material was evaluated, including its hysteresis expansion, hysteresis-loop parameters, α-Fe crystal formation, thermogravimetric transfer, and Curie temperature. The center point of the casing was subjected to flame local-heating annealing to attain ferromagnetism and paramagnetism material characteristics. The experimental shell was between these magnetic-phase-transition properties and was used to observe the magnetic power and noise characteristics of the microgenerator. The measured magnetic flux at the center of the amorphous shell was 1.2–2.4 mT, and the magnetic flux distributed around the shell was 0.6–1.0 mT. The generator with the amorphous-alloy shell had the lowest demagnetization rate in the permanent magnet region, which was close to the bottom of the pole piece, and the magnetic flux leakage of the pole-piece side frame changed the magnetic flux path, thus affecting the demagnetization performance. For the noise experiment, the flame-annealing temperature of the local center point of the amorphous casing reached the Curie temperature, and the noise characteristics of the casing can be reduced by 15 dB compared to those of the generator without the casing. However, the overall performance of generator harmonics and power were not fully improved.

## 1. Introduction

Modern high-speed-operation motors require long-lasting casings that experience low wear when driving at high speeds [1,2]. Consideration of housing features is indispensable in motor design, as they must be integrated with the rotor and stator to ensure high load-carrying capacity. Because of the complex electromagnetic-circuit structure of the motor, stable magnetic power, wide temperature regulation, and high motor stability are design priorities [3,4]; the effect of the motor casing and rotor stator on these properties must be considered. The dynamic analysis method is used to calculate the effects of the casing during rotation. The friction torque measured when the housing is working against the bearing can cause a torque imbalance. Installing a co-rotating sensor on the rotor enables calculation of the friction torque. When the static characteristics of the simulated housing and bearing are included, the coupling effect can be solved theoretically by the finite element method for multiple boundary conditions [5,6]. In the operation of a multi-degree-of-freedom motor with a housing bearing, the air film formed by the bearing is affected during the support process of the motor rotor and electromagnetic operation. Therefore, when the entire motor is energized, the electromagnetic excitation would cause different degrees of resonance in the motor, affecting the stability of the housing bearing [7,8].

Therefore, reducing the zero-mode vibration of permanent-magnet synchronous motors (PMSMs) is one of the most effective ways to meet the low vibration and noise requirements. Existing methods to reduce zero-mode vibration include electromagnetic optimization, mechanical design, and control strategies [9,10]. Electromagnetic optimization mainly aims to reduce harmonics in the magnetic field by optimizing the shape of the permanent magnets (PMs) and stator slots [11]. In the literature, the correlation between electrical harmonics and torque generated in the air gap and the overall enclosure has been discussed [12,13,14]. It has been proposed that a brushless PMSM drive coupled with a DC generator and a three-phase inverter can run synchronously; the harmonics of the high-voltage inverter can predict the torque vibration and the overall life of the housing. The output voltage and current of the three-phase inverter varies with high ripple harmonics, which will cause torque ripple in PMSM drives. Such ripples can have a considerable effect on the enclosure because of the abnormal torsional vibration and noise generated in the PMSM.

Further research on these soft magnetic materials has considered various modifications to the production process, and additives or grain orientation modification have resulted in improved performances and reduced core power losses of up to 5% during operation at 50 Hz [15,16]. The cuboid soft magnetic bulk core made of amorphous FeSiB strips with a high saturation induction of Bs = 1.56 T, satisfactory mechanical strength, and relatively low power loss can successfully replace conventional Fe–Si steel production of harmonic filters. These filters are used in the grid during the transfer of electrical energy from the source to the object of application (i.e., the load). Their mission is to suppress harmonic currents flowing into the power system from the source by reducing harmful current distortion caused by nonlinear loads. So far, there are not many publications on the application of amorphous and nanocrystalline soft magnetic materials in harmonic filter choke coils. The main part of the filter is the choke coil assembly, which consists of a specific inductor L and capacitor C to ensure the desired suppression of higher harmonics. Studies [17,18] show the effect of material type (Fe–Si steel with 3 wt% Si; FeSiB) on the choke core power loss. Their results show that these losses are 7.7 W/kg in Fe–Si steel with 3 wt% Si core and a mass of 18.5 kg at B = 0.8 T at a magnetizing field frequency of 350 Hz, while in ${\mathrm{Fe}}_{78}{\mathrm{Si}}_{13}B_{9}$ cores of the same mass these losses are reduced to 0.96 W/kg. A general trend in magnetic material selection for output filter chokes of high frequency inverters (up to 150 kHz) powered by renewable energy sources is presented. The literature [19] has demonstrated the magnetic properties of bulk cores made from rapidly quenched tapes cast from a newly developed soft magnetic nanocrystalline alloy with FeCuMoSiB composition. This study compared the soft magnetic properties of nanocrystalline Fe-based bulk cores with amorphous FeSiB bulk cores and found that at B = 1.5 T and f = 600 Hz the power loss of nanocrystalline and amorphous cores are 11 W/kg and 7 W/kg, respectively. This proves that nanocrystalline materials are not significantly better than amorphous materials under such operating conditions, especially in terms of magnetic field frequencies. In another study [20], the results of applying nanocrystalline bulk cores produced from Fe-based ribbons in high frequency choke coils (10 kHz) are presented. Magnetic cores produced from nanocrystalline ${\mathrm{Fe}}_{56}{\mathrm{Co}}_{24}{\mathrm{Nb}}_{4}B_{13}{\mathrm{Si}}_{2}{\mathrm{Cu}}_{1}$ ribbons for 25 kW DC–DC converters, rated to operate in the discontinuous conduction mode at 300 A peak current and switching frequencies of up to 20 kHz, have been described elsewhere [21,22].

However, in-depth research and discussion of the relationship between the generator casing and magnetic properties is lacking. In this study, a bipolar PM generator was developed to investigate whether coating the amorphous iron core could increase the magnetic characteristics and suppress the influence of noise and electromechanical vibration, as shown in Figure 1. An improved flux-enhanced PM generator was developed; unlike existing PM generators with iron/steel shells, the proposed generator has a shell of amorphous alloy. Moreover, it works at high temperature because the magnetic flux distribution is changed. The demagnetization phenomenon is the transient magnetic performance of a fault. This paper is organized as follows: Section 1 describes the application of amorphous materials. Section 2 details the properties of the amorphous material and the flame sintering process after annealing of the coated PM machine. The harmonics and losses of the motor are introduced. In addition to the noise and other characteristics, Section 3 presents a simulation of the shell under local flux for different annealing conditions. Section 4 describes the experiment setup and presents the experimental results. The entire study is summarized in Section 5.

## 2. Measurement and Analysis of Ferromagnetic and Paramagnetic Properties of Amorphous Alloy Shells

This study aimed to measure the magnetic properties of FeSiB ribbons at different annealing temperatures in order to further understand the correlation between the two through microstructure analysis and observation. Firstly, the thermal properties of the ribbons were analyzed by differential scanning calorimetry (DSC), then the phase composition of the annealed ribbons was identified by X-ray diffraction (XRD). Scanning electron microscopy (SEM), which uses a highly focused electron beam to scan the surface of the sample, was used to observe the microstructure of the thin strip section. The brittleness of the ribbon was determined by the bending test to determine the brittleness temperature of the ribbon after annealing, and the vibrating-sample magnetometer (VSM) and the magnetostriction meter were then used to measure the hysteresis curve of the ribbon and the amorphous-material length changes.

### 2.1. High Temperature Magnetic Transitions by DSC

By using a magnetic meter, the SA1 metal content (${\mathrm{Fe}}_{89.1}{\mathrm{Si}}_{7.4}B_{2.36}$) is measured by inductively coupled plasma mass spectrometry (ICP-MS). Thermogravimetric analysis (TGA) is a method of changing the physical and chemical properties of a substance through an increase in temperature (isothermal heating rate) or time (isothermal and/or mass conservation loss). A differential scanning calorimeter (MDSC 2920, TA Instruments, New Castle, DE, USA) was used to analyze 11-mg samples of SA1 amorphous alloy between 25 and 800 °C at a temperature increase rate of 5 °C/min. The DSC measurement results from this experiment are shown in Figure 2a. This iron-based amorphous alloy has a crystallization exothermic peak, and, when the heating rate increases, the crystallization peak shifts to a higher temperature. This shows that the amorphous-alloy material has a direct correlation between composition and crystallization behavior. As shown in Figure 2a, the exothermic peak is strongly influenced by the Fe and B contents: when these are higher, the crystallization peak also rises. The measurements were acquired at low temperature. The crystallization peak temperature of Fe-based FeSiB amorphous ribbon was 482.6 °C, and the crystallization activation energy was measured as 19.5 J/g.

### 2.2. Material Weight Change and Curie Temperature

An amorphous magnetic material must measure the Curie temperature ($T_{c}$) process of the phase change that reaches a specific temperature. Thermogravimetric analysis (TGA) was performed using a thermogravimetric analyzer to observe the weight change in the physical and chemical properties of the materials as the temperature (equal heating rate) or time (isothermal and mass conservation loss) increased, as shown in Figure 2b. The TGA test result is of SA1 at 385.6 °C. This result is the phase-transition process of the magnetic material and is more stable for annealing temperature processing. It indicates that ~360 °C is suitable for the annealing, owing to the $T_{c}$ effect.

### 2.3. Magnetic Properties of BH Curve for Amorphous Shells

After the annealing of the amorphous-alloy strips was completed, a VSM (VSM-C7, TOEI) was used to measure the hysteresis curve of the annealed ribbons (Figure 3a). The parameters $B_{s}$, $B_{r}$, and $H_{c}$ were obtained. Below the $T_{c}$, which lies between 340 and 360 °C, of the TGA measurement phase change, the reordering between atoms and the bonding began to approach each other. Meanwhile, the crystalline phase α-Fe appeared, and the reordered atomic structure further increased the $B_{s}$ value, while the $H_{c}$ value represented the difficulty of the magnetization process. Annealing treatment eliminated the residual stress in the thin strip, and the coercive force was the smallest, which affects the core loss. The hysteresis-loop measurement parameters were obtained. The magnetic squareness ratio coefficient of SA1 was found to be 0.007892. This shows that the noise value of the test core may have a proportional trend.

### 2.4. Magnetostriction and Magneto-Mechanical Vibration

Magnetostriction-measuring equipment (TLVM-AC, TOEI) was used to measure the magnetostriction. The magnetostriction characteristics selected for measurement were those suitable for soft magnetic materials, such as silicon steel sheets and amorphous alloys. During magnetization, mechanical contraction occurs in the magnetization direction, owing to the hysteresis. The measured magnetostriction (ppm) value of SA1 is 60 ppm, as shown in Figure 3b.

### 2.5. Relationship between Phase Crystallization Peaks and Hysteresis Loops

Figure 4 shows the results of XRD analysis of the phase structure of the ribbon before and after annealing. The figure indicates that the annealing temperature below 400° is still the same as the diffraction result of the unannealed thin strip, showing a broad peak of 2θ within 40–50°, which is evidently amorphous. Until the 400° point, a small diffraction peak of α-Fe (110) appears at 44.6°. After the annealing process, the crystalline state of the material was checked, and the properties of the material were observed using an X-ray diffractometer (D8 DISCOVER SSS Multi-Function High Power X-ray Diffractometer). The sharp peak in Figure 4a indicates that α-Fe especially exhibits strong crystallization between annealing temperatures from 360 °C to 450 °C. At a higher temperature of 450 °C, not only does the crystalline signal of α-Fe becomes stronger, but also that the diffraction peaks of crystalline phases such as ${\mathrm{Fe}}_{2}B$ and ${\mathrm{Fe}}_{3}B$ begin to appear. These results, in terms of the reduction of the saturation magnetic flux density and magnetic permeability of the magnetic material at 360 °C, are shown. However, unexpectedly, the vibration and noise between 390 °C and 450 °C were also reduced to a low level, owing to the deterioration of the magnetic properties. The rate of change in lagging expansion and contraction was evidently slower. Furthermore, the annealing temperature of the amorphous-alloy material is between 360 °C and 450 °C, and the hysteresis loop exhibits different changes. This XRD diffraction analysis result corresponds to the curve measured by the previous DSC. It can be inferred that the single peak in the DSC curve represents the superposition of two crystallization reactions. The crystallization reaction temperature ranges of Fe and $F_{x}B$ are remarkably close, and there is a possibility of overlap during the slow heating process that tends to be stable. However, the difference between the two crystallization reactions would be increasingly prominent under the condition of faster heating rate. This may explain the correlation between the two sets of data from XRD and DSC.

By measuring the VSM hysteresis curves at different annealing temperatures, different parameters such as saturation magnetization ($B_{s}$) and coercivity ($H_{c}$) can be obtained, as discussed in subsequent sections. The BH measurement results at different annealing temperatures are plotted in Figure 4b. The amorphous ribbon has higher B_s_ and lower $H_{c}$ when the annealing temperature at 360 °C is lower than $T_{c}$. On the contrary, if the annealing temperature is higher than Tc, the crystalline state will gradually appear. At this time, B_s_ decreases and $H_{c}$ increases. Increasing the annealing temperature causes the area of the B–H ring to become larger. The difference between the two can explain the property change caused by the structural change of the thin strip after heat treatment.

### 2.6. Material Brittle and Ductile Microstructure after Annealing

The specimen used in the experiment is a thin FeSiB amorphous ribbon with a thickness of about 25 μm. The raw material strips were annealed at 350, 390, and 450 °C for 120 min under nitrogen protection. The brittleness of the strip after the annealing treatment was determined using bending tests. The strips were cut into approximately to the size of 1 cm × 6 cm at room temperature, bent into a U-shape, and placed between two parallel plates. After the parallel plates were slowly pressed together, a reference for the cross-section of the thin strip of material could be observed after it became brittle. After pressing slowly, the distance d is measured between the two plates when the thin strip breaks. The reference strain $\varepsilon_{f}$ when the thin strip is broken can be estimated and calculated by the following simple formula as reference data for assessing the brittleness of the thin strip. The same test piece is tested five times to find the average rupture strain value, as follows:(1)$$\varepsilon_{f}=\frac{t}{d-t}$$
where t is the thickness of the thin strip, which is known to be about 25 μm. It is defined that when $\varepsilon_{f}$ < 1.0, the thin strip begins to appear embrittled, which can be used to determine the brittle-transition temperature of the thin strip after annealing.

The microstructure and material properties of the materials were obtained using SEM (JEOL, JSM-6500F). Different annealing temperatures were controlled between 350 and 450 °C for the ductile- and brittle-phase transitions, as shown in Figure 5a. As shown in Figure 5b, the SEM reveals the transition from ductile to brittle, where the temperature of the first stage is the optimal magnetic transition temperature below the Curie point temperature in the body-centered cubic (BCC) α-Fe. Then, in order to bending the material in half into a U-shape, it is about the cross-section formed by folding it in half by 0.2 mm. As shown in Figure 5c, the temperature of the second stage is increased slightly beyond the Curie point, the crystal material is gradually converted from α-Fe of BCC to FCC, and the material is bent into a U-shaped cross-section, formed by folding it in half to about 0.5 mm. The third stage is to increase the annealing temperature of the material to approximately the crystallization temperature, resulting in the complete FCC crystallization condition of the material. As shown in Figure 5d, the cross-section was formed by bending the material by about 10 mm, into a U-shape. The annealed iron ribbon gradually loses its magnetic properties and is almost crystalline, and the physical properties of hysteresis are also gradually lost. The measurement of the brittleness of the thin strip is carried out via the bending test.

At room temperature, the thin strip is cut into strips of about 1 cm × 5 cm, bent into a U-shape, and placed between two parallel plates. When the rupture strain of the ribbon is equal to 1, the ribbon is still in a ductile state, but when the rupture strain is less than 1, the ribbon becomes brittle. It can be seen from the figure that the temperature at which the thin strip starts to become brittle is between 360 °C and 390 °C. With the increase of the annealing temperature, the fracture strain decreases rapidly, and the deviation value is large at this time. When the annealing temperature is 450 °C, which is the crystallization temperature, the rupture occurs. The strain approaches zero and the deviation value approaches zero. The above change trend is also in line with the change trend of the electromagnetic properties measured previously, that is, the effect of annealing on the mechanical properties can be divided into two stages; the short-range ordered structure produced by low-temperature annealing causes the ribbon to become brittle. However, it is only partially brittle, which may be because of the difference between the stress and strain of the short-range ordered structure and the original disordered amorphous structure; when the difference is even greater, its interface becomes the fracture-inducing region of stress concentration.

## 3. Ferromagnetic Material Surface Local Annealing and Simulation Results

In this study, a sinusoidal waveform was generated between the surface and the air-gap magnetic flux density on the rotor core structure. The embedded PM core had mirror-symmetric magnetization directions. Generators with semi-magnetic materials were used to guide the path of the flux. The radial magnetic field was stronger because of the placement of the PMs. The angle between the two PMs at the left and right ends could change the demagnetization performance of the magnetic field in the magnetic pole. The two-pole AC generator used in this research produced changes in the magnetic field through rotation of the generator, producing an alternating current that can be converted into direct current through slip-rings, brushes, and commutators. Figure 6 shows the generator with amorphous-alloy and local annealing area sketch form. The rotating electromagnetic field maintained a fixed polarity, passed through wound coils embedded in the surrounding stator, and penetrated them as the rotating magnetic field of the rotor extended outward. Then, the magnetic flux passed through the generator winding and amorphous shell.

In this study, a single-phase PM generator was built. The rotor of the generator was a bipolar NS type, and the stator was a double-coil hollow which was non-magnetic material-clad structure. The rotor and the stator were integrated to be the generator type of single-phase structure. The magnetic flux was mapped to the generator shell. PMs with unipolar magnetic flux were used; the unilateral-magnetic-flux path behavior was obtained by PMs that provided varying magnetization in the lateral direction. Simulation results showed that a cylindrical single-sided permanent magnet and a thin sheet of high-permeability material can simulate a thin magnetic material, avoiding the difficulty of volumetric meshing of thin extended structures in 3D. Using the constitutive relation between the flux density and magnetic field
(2)$$B=\mu_{0}\left(H+M\right)$$
(3)$$-\nabla \cdot \left(\mu_{0}\nabla V_{m}-\mu_{0}M\right)=0$$
where $V_{m}$ is the magnetic potential, $\mu_{0}$ is the vacuum permeability, H is the magnetic flux strength, and M is the laterally periodic magnetization. The magnetic flux is parallel to the upper surface of the shell, and no magnetic flux appears above it. The laterally periodic magnetization only appears on one side of the magnet. On the other hand, the magnetic flux appears inside the magnet, generating a holding force that is two times stronger than that of a conventional magnet. Figure 7 shows that the calculated magnetic flux density and direction for the version of a one-sided magnet was obtained, including magnetic saturation in the amorphous alloy shell. The magnetic-flux-saturation effects induced a smaller force on the material surface. The comparison shows that the magnetic force annealing temperature (360 °C) is higher for the other case (450 °C) with one-sided magnetization when compared to a uniform magnetization with the same amplitude.

## 4. Experiment Result and Discussion

The experimental environment and setup for the generator was as follows: A thin amorphous shell was used as a generator casing, as shown in Figure 8. As described in Section 2, the amorphous materials were manufactured at different annealing temperatures. Therefore, the amorphous alloys in different shells had different magnetic responses, including magnetic-loss, harmonic, noise, and vibration characteristics. The two-pole generator, experimental setup, and initialization are shown in Figure 8. The equipment included an oscilloscope (TDS 220, Tektronix, Beaverton, OR, USA), flux meter (TM-801, Kanetec Co., Ltd., Tokyo, Japan), power analyzer (PW3336, Hioki, Kagoshima, Japan), vibration meter (1332B, Showa, Saitama, Japan), and noise meter (TM-101, Tenmars, Taipei, Taiwan).

An AC voltage was induced in the stator coils. Because the output power was generated by the stator winding, the output could be connected through fixed terminals. The advantage of this structure is that there were no sliding contacts: the entire output circuit was continuously insulated and therefore could handle a large current flow. However, a certain portion of the output power was lost because of the set shell-less structure, magnetic coating, and alternating phenomena. The experimental process was as follows: The single-phase two-pole generator produced an average AC output voltage of 37 V; current of 0.11 A; power, P, of 4.11 W; and reactive power, Q, of 0.48 VAR. The operating frequency was set at 30 Hz; the power factor was 0.99. S is 4.14 VA. The bare-air generators and the coated amorphous-shell structure were integrated into the complete generator coil and rotor. The magnetic characteristics were compared, and it was observed that the generator produced energy and a harmonic state. In practice, most of the harmonics are odd harmonics, usually 3rd, 5th, 7th, and 9th, while even harmonics are rare. The experiments revealed that voltage and current can be measured without interrupting the generator operation. From the perspective of total harmonic distortion (THD) performance, as shown in Figure 9, both the bare-air-type (no cover, NC) and shell-type (cover) generators have 3rd order harmonics; the shell of the generator is larger for 3rd order harmonics; for 5th and 7th order harmonics the shell is smaller.

Since the magnetic field of the motor had no mapping-penetration phenomenon, it could maintain a relatively complete magnetic-field line for magnetic alternation and power conversion. Local flame-type amorphous-shell manufacturing is illustrated in Figure 10. In the experimental process, the amorphous alloy was sintered with a high-temperature flame (at a temperature of more than 400 °C, which is higher than the temperature of the Curie crystalline phase transition). Tesla-meter measurements showed that the sintered center-point magnetic flux density was approximately 10–15 mT, which could reach ~20 mT, compared to the marginal non-sintered amorphous thin strip. This verifies that, when the material is annealed or sintered at higher temperatures, its magnetic properties deteriorate.

The noise characteristics of the generator are reflected by the amorphous shell. As shown in Figure 11, the shell structure covered the AC motor. Several types of structures were used to process annealing conditions. The first type was the amorphous shell without an annealing process; the noise value was only 50 dB. The second type was that covered with sheets after annealing treatment; the noise value went up to ~79 dB. The third type used a local center annealing process for the amorphous shell; the noise value was 65.4 dB. The fourth type was with an annealing process at over the Curie temperature; the lowest noise value was only about 48.3 dB. The fifth type was the generator without casing; the noise value approached ~52 dB. Because the local amorphous sheet could be changed from ductile to brittle form, when the temperature reached 360 °C the crystalline structure of α-Fe was produced. An annealing procedure with an annealing temperature greater than the Curie temperature has a noise difference of at least about 15 dB compared with partial annealing or complete annealing procedures. Therefore, for the fourth type of Tc (compared with second-type annealing process), the noise could be suppressed by approximately 30 dB, owing to the brittle structure characteristic.

Additionally, the magnetic flux density on the amorphous surface was reduced by approximately 20%. High-temperature flame annealing had poor results for the power and magnetic characteristics of the AC motors. In particular, if the material was processed via partial flame sintering, the probability of developing cracks and damage was high, owing to the structural brittleness; the magnetic alternating characteristics also deteriorated. The noise level did not change drastically because of the hysteresis vibration. In comparison with normal annealing and amorphous materials without the annealing process, the noise was lowered to ~50 dB.

## 5. Conclusions

This study considered a micro-alternator operated with different housing structures: an uncoated shell structure and a shell-coated generator made of an iron-based amorphous-alloy soft magnetic material. The electromagnetic power and noise characteristics of generators with different shell structures were measured and analyzed. The amorphous-alloy material SA1 was used for the shell coating. The experimental shell was between the above magnetic phase-transition characteristics, included as paramagnetic and ferromagnetic phases. The magnetic fluxes of the magnetic phase transitions of the amorphous shell were measured by a flux meter. The magnetic flux in the center of the amorphous shell was 1.2–2.4 mT, whereas the magnetic flux distributed around the shell was 0.6–1.0 mT. The flame-annealing temperature at the local center point of the amorphous casing reached the Curie temperature, and the noise of the casing could be reduced by at least 15 dB compared with that of the generator without casing. Additionally, for a higher Curie (Tc) shell (compared to the annealing shell) the reduction of noise was ~30 dB. However, the overall performance, in terms of harmonics and power, was not significantly improved in the generator with the amorphous shell. 

## Figures and Tables

**Figure 1 micromachines-13-02086-f001:**
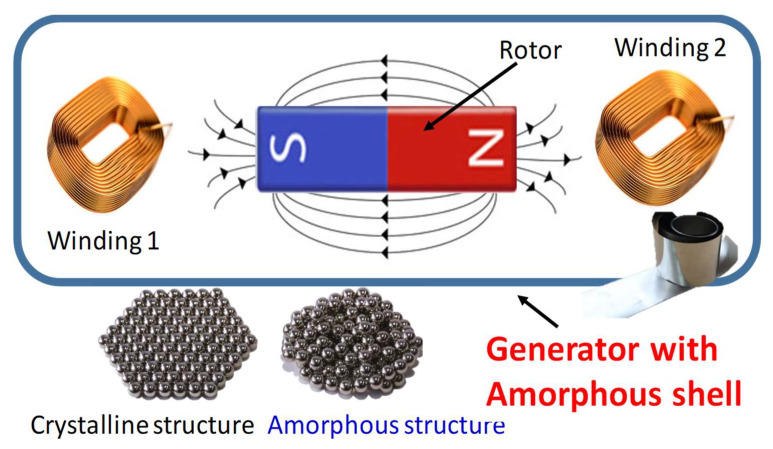
Single phase generator shell with amorphous alloy.

**Figure 2 micromachines-13-02086-f002:**
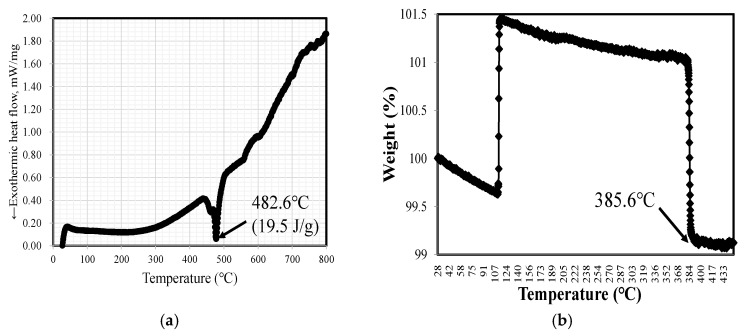
Magnetic properties of amorphous alloy: (**a**) exothermic heat flow by scanning calorimetry and (**b**) thermogravimetric analysis.

**Figure 3 micromachines-13-02086-f003:**
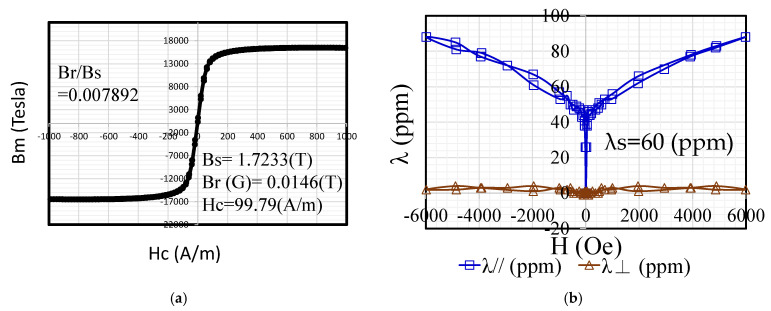
Magnetostriction measurement of amorphous SA1 material: (**a**) VSM and (**b**) magnetostriction.

**Figure 4 micromachines-13-02086-f004:**
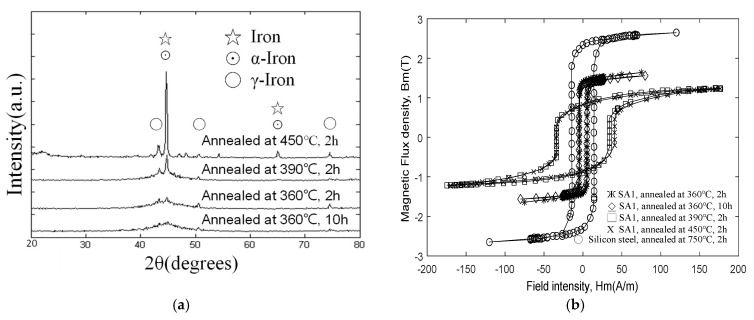
X-ray diffraction measurement results: (**a**) Sample SA1 crystallization peak at different temperatures and (**b**) Sample SA1 to crystallization peak and α-Fe at different temperatures.

**Figure 5 micromachines-13-02086-f005:**
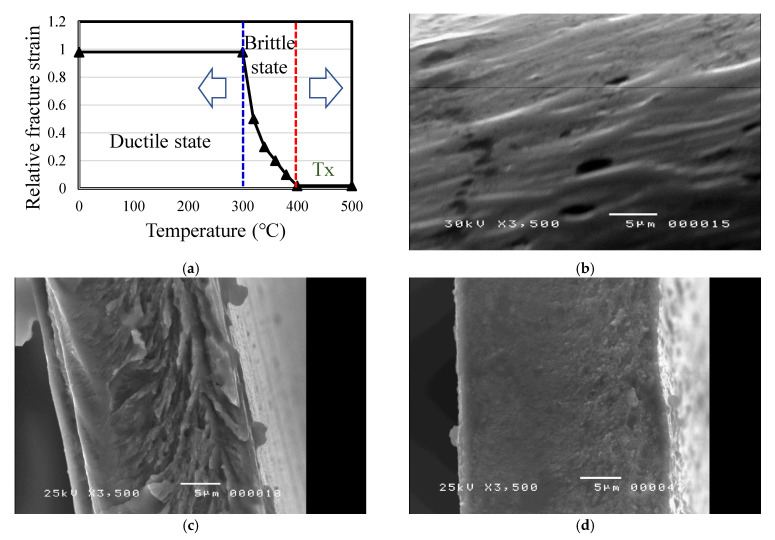
Measurement of the microstructure of material sample 3: (**a**) measurement of ductility and brittleness temperature range after annealing; SEM images at (**b**) 350 °C, (**c**) 390 °C, and (**d**) 450 °C.

**Figure 6 micromachines-13-02086-f006:**
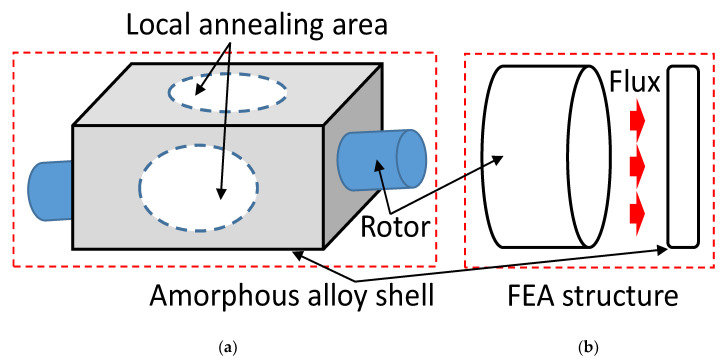
Experiment structure: (**a**) flux path on surface of the amorphous shell and (**b**) coated (uncoated) amorphous alloy of the generator shell.

**Figure 7 micromachines-13-02086-f007:**
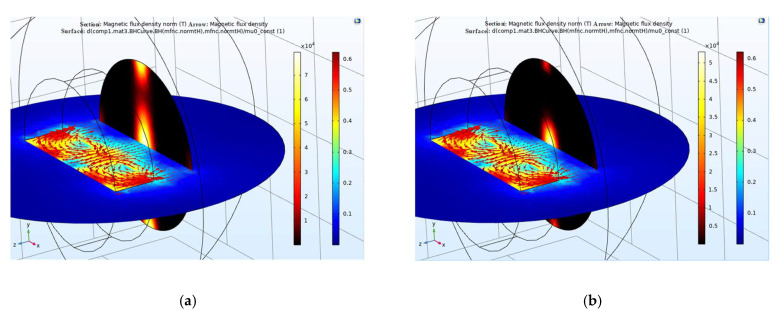
Amorphous-alloy shell depending on the flux variation: (**a**) annealing temperature 350 °C and (**b**) annealing temperature above Curie temperature at 450 °C.

**Figure 8 micromachines-13-02086-f008:**
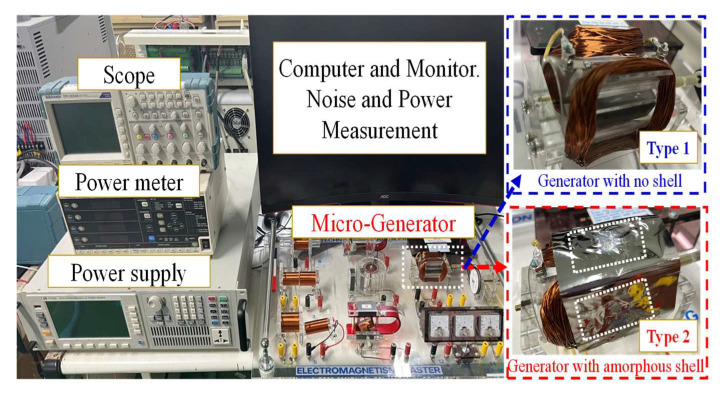
Experimental environment and setup.

**Figure 9 micromachines-13-02086-f009:**
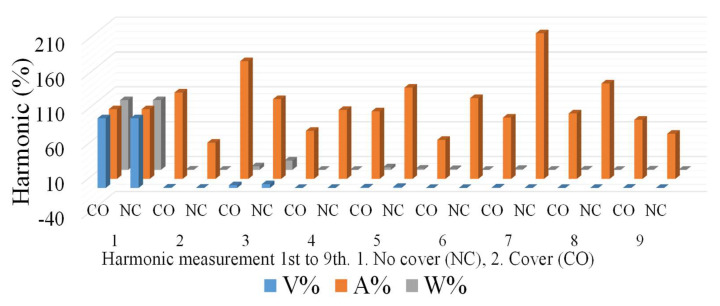
Harmonic performance experiment results of generator with or without amorphous shell.

**Figure 10 micromachines-13-02086-f010:**
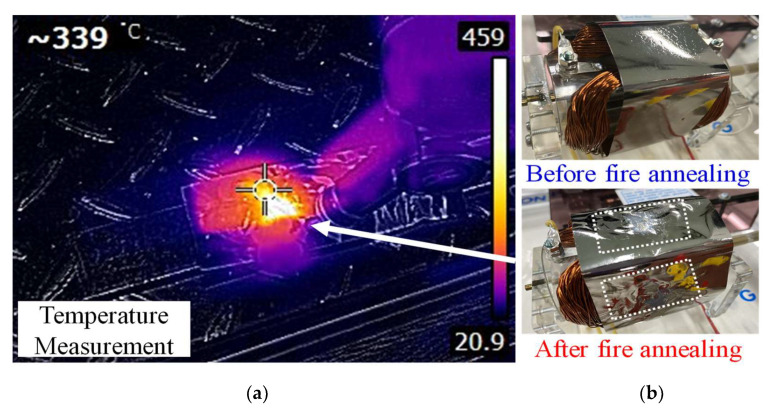
Experimental amorphous-material flame-annealing method used in the shell manufacturing process: (**a**) local annealing process; (**b**) differences in amorphous shell structure.

**Figure 11 micromachines-13-02086-f011:**
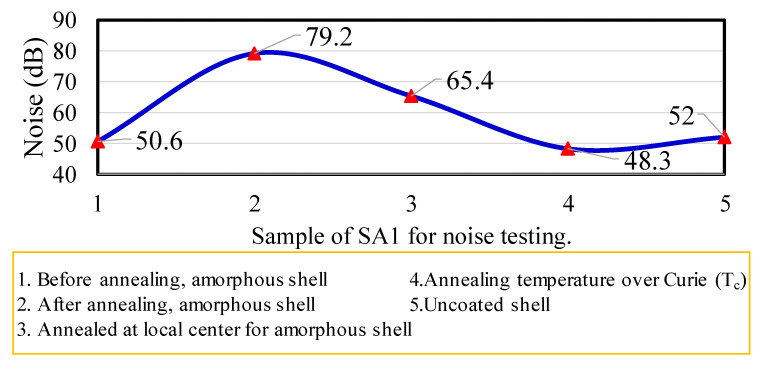
Measured generator noise with different annealing treatments of the amorphous shell.

## Data Availability

There is no data availability.

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
