# Peer review of "Effect of Local Annealing on Magnetic Flux Distribution and Noise in a Micro-Generator with Amorphous Shell"

_micromachines, 2022, doi:10.3390/mi13122086_

Round 1

Reviewer 1 Report

This manuscript analyzed the influence of annealing on magnetic field properties of a generator with amorphous alloy shell. However, for further review, English of this manuscript needs to be largely improved firstly. There are too many errors in the current manuscript and some of them affect the understanding of the context. In addition, this manuscript is illogical and the research purpose and significance are not clear. Please read some technical publications and learn how to write one, then go through your manuscript. It is suggested to rewritten your “Abstract, Keywords and Body”. Most importantly, the compliant positioning stage presented in this manuscript is not innovative and scientific. Overall, I don’t recommend this paper for publication in its current state.

Some specific errors:

(1) In Abstract, what’s the meaning of this sentence “Different permanent magnet motors have …”? “Different permanent magnet motors” means?

(2) In Abstract, the author says the flame annealing process could help reduce noise by 5 dB but the overall performance is unsatisfactory. This is contrary to the conclusion in Section 5. Which conclusion is right and which influence is more important?

(3) In Introduction, there is a grammatical error in the sentence “The conclusion summarized in section V”.

(4) Most of the charts in this manuscript are not clear and they look like screenshots. The authenticity of these charts is questionable.

......

Author Response

Review Reply

2022/11/2

Title: Effect of Magnetic Flux Distribution and Noise of Local Annealing for Micro-Generator with Amorphous Shell

Manuscript ID: micromachines-1962722

Authors: Chang-Hung Hsu *Received: 25 September 2022E-mails: chshiu@mail.aeust.edu.tw

Submitted to section: E:Engineering and Technology,

Dear Editor,

Q1: Please check that all references are relevant to the contents of the manuscript.

Reply: Thank you very much for valuable comments. This paper has been cited the relevant paper in this study, as shown in reference (Highlight with celadon color)

Q2: Any revisions to the manuscript should be highlighted such that any changes can be easily viewed by the editors and reviewers.

Reply: Thank you very much for valuable comments.

Q3: Please provide a cover letter to explain, point by point, the details of the revisions to the manuscript and your responses to the referees’ comments.

Reply: Thank you very much for valuable comments.

Q4: if you found it impossible to address certain comments in the review reports, please include an explanation in your appeal.

Reply: Thank you very much for valuable comments.

Q5: The revised version will be sent to the editors and reviewers.

Reply: Thank you very much for valuable comments.

Dear Reviewer 1

Comments and Suggestions for Authors

Q1: This manuscript analyzed the influence of annealing on magnetic field properties of a generator with amorphous alloy shell. However, for further review, English of this manuscript needs to be largely improved firstly. There are too many errors in the current manuscript and some of them affect the understanding of the context. In addition, this manuscript is illogical and the research purpose and significance are not clear. Please read some technical publications and learn how to write one, then go through your manuscript. It is suggested to rewritten your “Abstract, Keywords and Body”. Most importantly, the compliant positioning stage presented in this manuscript is not innovative and scientific. Overall, I don’t recommend this paper for publication in its current state.

Reply: Thank you very much for valuable comments. This study has been revised by native speaker, as shown up-load all of revision document.

Some specific errors:

  • In Abstract, what’s the meaning of this sentence “Different permanent magnet motors have …”? “Different permanent magnet motors” means?

Reply: Thank you very much for valuable comments. It has been revised abstract as below. (Highlight with yellow color)

Abstract: This study examines micro-alternators with two different housing structures: an uncoated shell and a shell coated with an iron-based amorphous-alloy soft magnetic material. The electromagnetic power and noise characteristics of generators with these shell structures are measured and analyzed. The material used for the shell coating is the SA1 amorphous alloy. The magnetic property of the SA1 material is evaluated, including their hysteresis expansion, hysteresis-loop parameters, α-Fe crystal formation, thermogravimetric transfer, and Curie temperature. The center point of the casing is subjected to flame local-heating annealing to achieve ferromagnetism and paramagnetism material characteristics. The experimental shell is between these magnetic-phase-transition properties and is used to observe the magnetic power and noise characteristics of the micro-generator. The measured magnetic flux at the center of the amorphous shell is 1.2–2.4 mT, and the magnetic flux distributed around the shell is 0.6–1.0 mT. The generator with amorphous alloy shell has the lowest demagnetization rate in the permanent magnet region, which is close to the bottom of the pole piece, and the magnetic flux leakage of the pole-piece side frame changes the magnetic flux path, thus affecting the demagnetization performance. For the noise experiment, the flame annealing temperature of the local center point of the amorphous casing reaches the Curie temperature, and the noise characteristics of the casing can be reduced by 15 dB compared to those of the generator without the casing. However, the overall performance of generator harmonics and power is not fully improved.

  • In Abstract, the author says the flame annealing process could help reduce noise by 5 dB but the overall performance is unsatisfactory. This is contrary to the conclusion in Section 5. Which conclusion is right and which influence is more important?

Reply: Thank you very much for valuable comments. The abstract has been revised as shown in below.

5Conclusion

This study considered a micro-alternator operated with different housing structures: an uncoated shell structure and a shell-coated generator made of an iron-based amorphous-alloy soft magnetic material. The electromagnetic power and noise characteristics of generators with different shell structures were measured and analyzed. The amorphous-alloy materials SA1 was used for the shell coating. The experimental shell was between the above magnetic phase-transition characteristics, included as paramagnetic and ferromagnetic phases. The magnetic fluxes of the magnetic phase transitions of the amorphous shell were measured by a flux meter. The magnetic flux in the center of the amorphous shell was 1.2–2.4 mT, whereas the magnetic flux distributed around the shell was 0.6–1.0 mT. The flame annealing temperature at the local center point of the amorphous casing reached the Curie temperature, and the noise of the casing could be reduced by at least 15 dB compared with that of the non-casing generator. Besides, a higher Curie (Tc) shell compare to the annealing shell, the reduction of the noise is around 30 dB. However, the overall performance of generator with amorphous shell for harmonics and power, was not fully improved.

  • In Introduction, there is a grammatical error in the sentence “The conclusion summarized in section V”.

Reply: Thank you very much for valuable comments. This study has been revised.

  • Most of the charts in this manuscript are not clear and they look like screenshots. The authenticity of these charts is questionable.

Reply: Thank you very much for valuable comments. All figure has been changed and revised. (Highlight with green color)

(a)                                   (b)

Figure 2. Detection of amorphous alloys in three samples using (a) differential scanning calorimetry and (b) thermogravimetric analysis.

(a)                                      (b)

Figure 3. Magnetostriction measurement of amorphous SA1 material: (a) VSM and (b) magnetostriction.

(a)                                      (b)

Figure 4. X-ray diffraction measurement results: (a) Sample SA1 crystallization peak at different temperatures and (b) Sample SA1 to crystallization peak and α-Fe at different temperatures.

(a)                                      (b)

(c)                                      (d)

Figure 5. Measurement of the microstructure of material sample 3: (a) Measurement of ductility and brittleness temperature range after annealing; SEM images at (b) 350 °C, (c) 390 °C, and (d) 450 °C.

Dear Reviewer 2

Comments and Suggestions for Authors

Q1: What is the novelty of the following work?

Reply: Thank you very much for valuable comments. The novelty and contribution are revised in Abstract and conclusion. (Highlight with yellow color)

Abstract: This study examines micro-alternators with two different housing structures: an uncoated shell and a shell coated with an iron-based amorphous-alloy soft magnetic material. The electromagnetic power and noise characteristics of generators with these shell structures are measured and analyzed. The material used for the shell coating is the SA1 amorphous alloy. The magnetic property of the SA1 material is evaluated, including their hysteresis expansion, hysteresis-loop parameters, α-Fe crystal formation, thermogravimetric transfer, and Curie temperature. The center point of the casing is subjected to flame local-heating annealing to achieve ferromagnetism and paramagnetism material characteristics. The experimental shell is between these magnetic-phase-transition properties and is used to observe the magnetic power and noise characteristics of the micro-generator. The measured magnetic flux at the center of the amorphous shell is 1.2–2.4 mT, and the magnetic flux distributed around the shell is 0.6–1.0 mT. The generator with amorphous alloy shell has the lowest demagnetization rate in the permanent magnet region, which is close to the bottom of the pole piece, and the magnetic flux leakage of the pole-piece side frame changes the magnetic flux path, thus affecting the demagnetization performance. For the noise experiment, the flame annealing temperature of the local center point of the amorphous casing reaches the Curie temperature, and the noise characteristics of the casing can be reduced by 15 dB compared to those of the generator without the casing. However, the overall performance of generator harmonics and power is not fully improved.

Q2: The introduction part needs more improvement with recent references such as:

[1] Zaitsev, I.; Bereznychenko, V.; Bajaj, M.; Taha, I.B.M.; Belkhier, Y.; Titko, V.; Kamel, S. Calculation of Capacitive-Based Sensors of Rotating Shaft Vibration for Fault Diagnostic Systems of Powerful Generators. Sensors 2022, 22, 1634. https://doi.org/10.3390/s22041634.

[2] Szabó, Z.; Fiala, P.; Zukal, J.; Dědková, J.; Dohnal, P. Optimal Structural Design of a Magnetic Circuit for Vibration Harvesters Applicable in MEMS. Symmetry 2020, 12, 110. https://doi.org/10.3390/sym12010110.

Reply: Thank you very much for valuable comments. All figure has been added and revised, both of 3 and 6.(Highlight with celadon color)

References

  1. Qiao, C. Jiang, Y. Zhu, and G. Li, “Research on design method and electromagnetic vibration of six-phase fractional-slot concentrated winding PM motor suitable for ship propulsion,'' IEEE Access, vol. 4, pp. 8535-8543, 2016.
  2. Kong, Z. Shuai, W. Li, and D. Wang, “Electromagnetic vibration characteristics analysis of a squirrel-cage induction motor under different loading conditions,'' IEEE Access, vol. 7, pp. 173240-173248, 2019.
  3. Zaitsev, V. Bereznychenko, M. Bajaj, I.B.M. Taha, Y. Belkhier, V. Titko, S. Kamel, “Calculation of Capacitive-Based Sensors of Rotating Shaft Vibration for Fault Diagnostic Systems of Powerful Generators,'' Sensors, vol. 22, pp.1634, 2022.
  4. G. Sarigiannidis and A. G. Kladas, “Switching frequency impact on permanent magnet motors drive system for electric actuation applications,” IEEE Trans. Magnetics, vol. 51, no. 3, 2015.
  5. Szabó, P. Fiala, J. Zukal, J. Dědková, P. Dohnal, “Optimal Structural Design of a Magnetic Circuit for Vibration Harvesters Applicable in MEMS. Symmetry, vol. 12, pp. 110, 2020.
  6. Zheng Sun et.as., Field programmable gate array-based torque predictive control for permanent magnet servo motors, Micromachines, vol. 13, no. 7, pp. 1055, 2022.
  7. Nagalingam Rajeswaran et. al., FPGA implementation of AI-based inverter IGBT open circuit fault diagnosis of induction motor drives,” Micromachines, vol. 13, no. 5, pp. 663-1-11. 2022.
  8. Rodrigo Hernandez-Alvarado et. al., Self-tuning control using an online-trained neural network to position a linear actuator,” Micromachines, vol. 13, no. 5, pp. 669-1-20, 2022.
  9. Gaur, ‘‘A new stress-based approach for nonlinear finite element analysis,’’ J. Appl. Comput. Mech., vol. 5, no. 3, pp. 563–576, 2019.
  10. Yin, X. Zhang, F. Ma, C. Gu, H. Gao, and Y. Wang, ‘‘New equivalent model and modal analysis of stator core-winding system of permanent magnet motor with concentrated winding,’’ IEEE Access, vol. 8, pp. 78140–78150, 2020.
  11. K. Moayyedi, ‘‘Extension ability of reduced order model of unsteady incompressible flows using a combination of POD and Fourier modes,’’ J. Appl. Comput. Mech., vol. 5, no. 1, pp. 1–12, 2019.
  12. X. Feng, ‘‘The application of mode synthesis method in the dynamic calculation of body structure,’’ Automot. Eng., vol. 34, no. 9, pp. 811–815, 2012.
  13. M. Marashi, “Estimating the mode shapes of a bridge using short time transmissibility measurement from a passing vehicle,'' J. Appl. Comput. Mech., vol. 5, no. 4, pp. 735-748, 2019.
  14. Y. Kim, J. K. Nam, and G. H. Jang, “Reduction of magnetically induced vibration of a spoke-type IPM motor using magnetomechanical coupled analysis and optimization,” IEEE Trans. Magnetics, vol. 49, no. 9, pp. 5097–5105, 2013.
  15. Ebrahimi, Y. Gao, H. Dozono, and K. Muramatrsu, “Comparison of time integration methods in magnetomechanical problems,” IEEE Trans. Magnetics, vol. 51, no. 3, 2015.
  16. Zheng Li, Structural design and analysis of hybrid drive multi-degree-of-freedom motor, Micromachines, vol. 13, no. 6, pp.955, 2022.
  17. Zheng Li et. al., Position detection method of piezoelectric driven spherical motor based on laser detection,” Micromachines, vol. 13, no. 5, pp. 662-1-11. 2022.
  18. Bao, E. W. Chen, Y. M. Lu, Z. S. Liu, and S. Liu, ‘‘Vibration and noise analysis for a motor of pure electric vehicle,’’ Adv. Mater. Res., vols. 915–916, pp. 98–102, 2014.
  19. S. Rahman, ‘‘Modified multi-level residue harmonic balance method for solving nonlinear vibration problem of beam resting on nonlinear elastic foundation,’’ J. Appl. Comput. Mech., vol. 5, no. 4, pp. 627–638, 2019.
  20. Hieu, ‘‘Free vibration analysis of quintic nonlinear beams using equivalent linearization method with a weighted averaging,’’ J. Appl. Comput. Mech., vol. 5, no. 1, pp. 46-57, 2019.
  21. Xie, C. Pi, and Z. Li, “Study on design and vibration reduction optimization of high starting torque induction motor,'' Energies, vol. 12, no. 7, pp. 1263, Apr. 2019.
  22. Ebrahimi, Y. Gao, A. Kameari, H. Dozono, and K. Muramatsu, “Coupled magneto-mechanical analysis considering permeability variation by stress due to both magnetostriction and electromagnetism,” IEEE Trans. Magnetics, vol. 45, no. 5, pp. 1621–1624, 2013.
  23. Y. Kim, G. H. Jang, and J. K. Nam, “Magnetically induced vibrations in an IPM motor due to distorted magnetic forces arising from flux weakening control,” IEEE Trans. Magnetics, vol. 49, no. 7, pp. 3929–3932, 2013.
  24. Suzana Uran,Božidar Bratina andRiko Šafarič, A microfluidic rotational motor driven by circular vibrations, Micromachines, vol. 10, no. 12, pp. 809, 2019.
  25. Wenxing Chen, Shuyang Dai and Baojuan Zheng, “A dynamic thermal-mechanical coupling numerical model to solve the deformation and thermal diffusion of plates,” Micromachines, vol. 13, no. 4, pp. 753-1-36. 2022.
  26. Belahcen, D. Singh, P. Rasilo, F. Martin, S. G. Ghalamestani, and L. Vandevelde, “Anisotropic and strain-dependent model of magnetostriction in electrical steel sheets,” IEEE Trans. Magnetics, vol. 51, no. 3, Mar. 2015.
  27. Oka, T. Ogasawara, N. Kawano, and M. Enokizono, “Estimation of suppressed iron loss by stress-relief annealing in an actual induction motor stator core by using the excitation inner core method,” IEEE Trans. Magnetics, vol. 50, no. 11, pp. 1–4, Nov. 2014.
  28. Chai, P. Liang, Y. Pei, and S. Cheng, “Analytical method for iron losses reduction in interior permanent magnet synchronous motor,” IEEE Trans. Magnetics, vol. 51, no. 11, 2015.
  29. -J. Kim, D.-Y. Kim, and J.-P. Hong, “Structure of concentrated flux type interior permanent-magnet synchronous motors using ferrite permanent magnets,” IEEE Trans. Magnetics, vol. 50, no. 11, 2014.

Q3: The conclusion needs to be improved with important results.

Reply: Thank you very much for valuable comments. This study novelty work is revised in conclusion.

5.  Conclusion

This study considered a micro-alternator operated with different housing structures: an uncoated shell structure and a shell-coated generator made of an iron-based amorphous-alloy soft magnetic material. The electromagnetic power and noise characteristics of generators with different shell structures were measured and analyzed. The amorphous-alloy materials SA1 was used for the shell coating. The experimental shell was between the above magnetic phase-transition characteristics, included as paramagnetic and ferromagnetic phases. The magnetic fluxes of the magnetic phase transitions of the amorphous shell were measured by a flux meter. The magnetic flux in the center of the amorphous shell was 1.2–2.4 mT, whereas the magnetic flux distributed around the shell was 0.6–1.0 mT. The flame annealing temperature at the local center point of the amorphous casing reached the Curie temperature, and the noise of the casing could be reduced by at least 15 dB compared with that of the non-casing generator. Besides, a higher Curie (Tc) shell compare to the annealing shell, the reduction of the noise is around 30 dB. However, the overall performance of generator with amorphous shell for harmonics and power, was not fully improved.

Dear Reviewer 3

Comments and Suggestions for Authors

Dear,

Q1: Authors need to make clear in the abstract what the problem is and the proposed contribution to solve it. It is not common to use Figures in Introduction. In this section, authors must contextualize recent articles on the topic and that show the reason for the proposed article. The article has only 13 references and needs improvement in this regard.

Reply: Thank you very much for valuable comments. The novelty and contribution are revised in Abstract and conclusion. (Highlight with yellow color)

Abstract: This study examines micro-alternators with two different housing structures: an uncoated shell and a shell coated with an iron-based amorphous-alloy soft magnetic material. The electromagnetic power and noise characteristics of generators with these shell structures are measured and analyzed. The material used for the shell coating is the SA1 amorphous alloy. The magnetic property of the SA1 material is evaluated, including their hysteresis expansion, hysteresis-loop parameters, α-Fe crystal formation, thermogravimetric transfer, and Curie temperature. The center point of the casing is subjected to flame local-heating annealing to achieve ferromagnetism and paramagnetism material characteristics. The experimental shell is between these magnetic-phase-transition properties and is used to observe the magnetic power and noise characteristics of the micro-generator. The measured magnetic flux at the center of the amorphous shell is 1.2–2.4 mT, and the magnetic flux distributed around the shell is 0.6–1.0 mT. The generator with amorphous alloy shell has the lowest demagnetization rate in the permanent magnet region, which is close to the bottom of the pole piece, and the magnetic flux leakage of the pole-piece side frame changes the magnetic flux path, thus affecting the demagnetization performance. For the noise experiment, the flame annealing temperature of the local center point of the amorphous casing reaches the Curie temperature, and the noise characteristics of the casing can be reduced by 15 dB compared to those of the generator without the casing. However, the overall performance of generator harmonics and power is not fully improved.

Reply: Thank you very much for valuable comments. All figure has been added and revised, both of 3 and 6.(Highlight with celadon color)

References

  1. Qiao, C. Jiang, Y. Zhu, and G. Li, “Research on design method and electromagnetic vibration of six-phase fractional-slot concentrated winding PM motor suitable for ship propulsion,'' IEEE Access, vol. 4, pp. 8535-8543, 2016.
  2. Kong, Z. Shuai, W. Li, and D. Wang, “Electromagnetic vibration characteristics analysis of a squirrel-cage induction motor under different loading conditions,'' IEEE Access, vol. 7, pp. 173240-173248, 2019.
  3. Zaitsev, V. Bereznychenko, M. Bajaj, I.B.M. Taha, Y. Belkhier, V. Titko, S. Kamel, “Calculation of Capacitive-Based Sensors of Rotating Shaft Vibration for Fault Diagnostic Systems of Powerful Generators,'' Sensors, vol. 22, pp.1634, 2022.
  4. G. Sarigiannidis and A. G. Kladas, “Switching frequency impact on permanent magnet motors drive system for electric actuation applications,” IEEE Trans. Magnetics, vol. 51, no. 3, 2015.
  5. Szabó, P. Fiala, J. Zukal, J. Dědková, P. Dohnal, “Optimal Structural Design of a Magnetic Circuit for Vibration Harvesters Applicable in MEMS. Symmetry, vol. 12, pp. 110, 2020.
  6. Zheng Sun et.as., Field programmable gate array-based torque predictive control for permanent magnet servo motors, Micromachines, vol. 13, no. 7, pp. 1055, 2022.
  7. Nagalingam Rajeswaran et. al., FPGA implementation of AI-based inverter IGBT open circuit fault diagnosis of induction motor drives,” Micromachines, vol. 13, no. 5, pp. 663-1-11. 2022.
  8. Rodrigo Hernandez-Alvarado et. al., Self-tuning control using an online-trained neural network to position a linear actuator,” Micromachines, vol. 13, no. 5, pp. 669-1-20, 2022.
  9. Gaur, ‘‘A new stress-based approach for nonlinear finite element analysis,’’ J. Appl. Comput. Mech., vol. 5, no. 3, pp. 563–576, 2019.
  10. Yin, X. Zhang, F. Ma, C. Gu, H. Gao, and Y. Wang, ‘‘New equivalent model and modal analysis of stator core-winding system of permanent magnet motor with concentrated winding,’’ IEEE Access, vol. 8, pp. 78140–78150, 2020.
  11. K. Moayyedi, ‘‘Extension ability of reduced order model of unsteady incompressible flows using a combination of POD and Fourier modes,’’ J. Appl. Comput. Mech., vol. 5, no. 1, pp. 1–12, 2019.
  12. X. Feng, ‘‘The application of mode synthesis method in the dynamic calculation of body structure,’’ Automot. Eng., vol. 34, no. 9, pp. 811–815, 2012.
  13. M. Marashi, “Estimating the mode shapes of a bridge using short time transmissibility measurement from a passing vehicle,'' J. Appl. Comput. Mech., vol. 5, no. 4, pp. 735-748, 2019.
  14. Y. Kim, J. K. Nam, and G. H. Jang, “Reduction of magnetically induced vibration of a spoke-type IPM motor using magnetomechanical coupled analysis and optimization,” IEEE Trans. Magnetics, vol. 49, no. 9, pp. 5097–5105, 2013.
  15. Ebrahimi, Y. Gao, H. Dozono, and K. Muramatrsu, “Comparison of time integration methods in magnetomechanical problems,” IEEE Trans. Magnetics, vol. 51, no. 3, 2015.
  16. Zheng Li, Structural design and analysis of hybrid drive multi-degree-of-freedom motor, Micromachines, vol. 13, no. 6, pp.955, 2022.
  17. Zheng Li et. al., Position detection method of piezoelectric driven spherical motor based on laser detection,” Micromachines, vol. 13, no. 5, pp. 662-1-11. 2022.
  18. Bao, E. W. Chen, Y. M. Lu, Z. S. Liu, and S. Liu, ‘‘Vibration and noise analysis for a motor of pure electric vehicle,’’ Adv. Mater. Res., vols. 915–916, pp. 98–102, 2014.
  19. S. Rahman, ‘‘Modified multi-level residue harmonic balance method for solving nonlinear vibration problem of beam resting on nonlinear elastic foundation,’’ J. Appl. Comput. Mech., vol. 5, no. 4, pp. 627–638, 2019.
  20. Hieu, ‘‘Free vibration analysis of quintic nonlinear beams using equivalent linearization method with a weighted averaging,’’ J. Appl. Comput. Mech., vol. 5, no. 1, pp. 46-57, 2019.
  21. Xie, C. Pi, and Z. Li, “Study on design and vibration reduction optimization of high starting torque induction motor,'' Energies, vol. 12, no. 7, pp. 1263, Apr. 2019.
  22. Ebrahimi, Y. Gao, A. Kameari, H. Dozono, and K. Muramatsu, “Coupled magneto-mechanical analysis considering permeability variation by stress due to both magnetostriction and electromagnetism,” IEEE Trans. Magnetics, vol. 45, no. 5, pp. 1621–1624, 2013.
  23. Y. Kim, G. H. Jang, and J. K. Nam, “Magnetically induced vibrations in an IPM motor due to distorted magnetic forces arising from flux weakening control,” IEEE Trans. Magnetics, vol. 49, no. 7, pp. 3929–3932, 2013.
  24. Suzana Uran,Božidar Bratina andRiko Šafarič, A microfluidic rotational motor driven by circular vibrations, Micromachines, vol. 10, no. 12, pp. 809, 2019.
  25. Wenxing Chen, Shuyang Dai and Baojuan Zheng, “A dynamic thermal-mechanical coupling numerical model to solve the deformation and thermal diffusion of plates,” Micromachines, vol. 13, no. 4, pp. 753-1-36. 2022.
  26. Belahcen, D. Singh, P. Rasilo, F. Martin, S. G. Ghalamestani, and L. Vandevelde, “Anisotropic and strain-dependent model of magnetostriction in electrical steel sheets,” IEEE Trans. Magnetics, vol. 51, no. 3, Mar. 2015.
  27. Oka, T. Ogasawara, N. Kawano, and M. Enokizono, “Estimation of suppressed iron loss by stress-relief annealing in an actual induction motor stator core by using the excitation inner core method,” IEEE Trans. Magnetics, vol. 50, no. 11, pp. 1–4, Nov. 2014.
  28. Chai, P. Liang, Y. Pei, and S. Cheng, “Analytical method for iron losses reduction in interior permanent magnet synchronous motor,” IEEE Trans. Magnetics, vol. 51, no. 11, 2015.
  29. -J. Kim, D.-Y. Kim, and J.-P. Hong, “Structure of concentrated flux type interior permanent-magnet synchronous motors using ferrite permanent magnets,” IEEE Trans. Magnetics, vol. 50, no. 11, 2014.

Q2: Figures 2 to 5 have poor resolution.

Reply: Thank you very much for valuable comments. The Figures 2 to 5 has been revised.

Reply: Thank you very much for valuable comments. All figure has been changed and revised. (Highlight with green color)

(a)                                   (b)

Figure 2. Detection of amorphous alloys in three samples using (a) differential scanning calorimetry and (b) thermogravimetric analysis.

(a)                                      (b)

Figure 3. Magnetostriction measurement of amorphous SA1 material: (a) VSM and (b) magnetostriction.

(a)                                      (b)

Figure 4. X-ray diffraction measurement results: (a) Sample SA1 crystallization peak at different temperatures and (b) Sample SA1 to crystallization peak and α-Fe at different temperatures.

(a)                                      (b)

(c)                                      (d)

Figure 5. Measurement of the microstructure of material sample 3: (a) Measurement of ductility and brittleness temperature range after annealing; SEM images at (b) 350 °C, (c) 390 °C, and (d) 450 °C.

Q3: The reviewer suggests removing the commercial from the Figures. For example, FLIR.

Reply: Thank you very much for valuable comments. The remark as FLIR has been removed.

(a)                           (b)

Figure 10. Experimental amorphous-material flame-annealing method used in the shell manufacturing process: (a) local annealing process; (b) differences in amorphous shell structure.

Q4: The conclusion section is very vague and inconsistent.

Reply: Thank you very much for valuable comments. The conclusion has been revised. (Highlight with yellow color)

5.  Conclusion

This study considered a micro-alternator operated with different housing structures: an uncoated shell structure and a shell-coated generator made of an iron-based amorphous-alloy soft magnetic material. The electromagnetic power and noise characteristics of generators with different shell structures were measured and analyzed. The amorphous-alloy materials SA1 was used for the shell coating. The experimental shell was between the above magnetic phase-transition characteristics, included as paramagnetic and ferromagnetic phases. The magnetic fluxes of the magnetic phase transitions of the amorphous shell were measured by a flux meter. The magnetic flux in the center of the amorphous shell was 1.2–2.4 mT, whereas the magnetic flux distributed around the shell was 0.6–1.0 mT. The flame annealing temperature at the local center point of the amorphous casing reached the Curie temperature, and the noise of the casing could be reduced by at least 15 dB compared with that of the non-casing generator. Besides, a higher Curie (Tc) shell compare to the annealing shell, the reduction of the noise is around 30 dB. However, the overall performance of generator with amorphous shell for harmonics and power, was not fully improved.

Reviewer 2 Report

What is the novelty of the following work?

The introduction part needs more improvement with recent references such as:

[1] Zaitsev, I.; Bereznychenko, V.; Bajaj, M.; Taha, I.B.M.; Belkhier, Y.; Titko, V.; Kamel, S. Calculation of Capacitive-Based Sensors of Rotating Shaft Vibration for Fault Diagnostic Systems of Powerful Generators. Sensors 202222, 1634. https://doi.org/10.3390/s22041634.

[2] Szabó, Z.; Fiala, P.; Zukal, J.; Dědková, J.; Dohnal, P. Optimal Structural Design of a Magnetic Circuit for Vibration Harvesters Applicable in MEMS. Symmetry 202012, 110. https://doi.org/10.3390/sym12010110.

The conclusion needs to be improved with important results.

Author Response

(The authors gave the same response as above.)

Reviewer 3 Report

Dear,

Authors need to make clear in the abstract what the problem is and the proposed contribution to solve it.

It is not common to use Figures in Introduction. In this section, authors must contextualize recent articles on the topic and that show the reason for the proposed article. The article has only 13 references and needs improvement in this regard.

Figures 2 to 5 have poor resolution.

The reviewer suggests removing the commercial from the Figures. For example, FLIR.

The conclusion section is very vague and inconsistent.

Author Response

(The authors gave the same response as above.)

Round 2

Reviewer 1 Report

The authors focus the paper on magnetic field properties of a generator with amorphous alloy shell. The topic is of interest. However, there exist some deficiencies in the papers. The main comments are listed as follow.

1.     The literature review is not enough, the conclusion of the review is quite rough and the authors' contributions can't be supported. Most of introduction underlines the torque, but it is unrelated to the main contents.

2.     In page 2, line 12, “In the literature, an LSM…..” , which literature? It lacks the necessary references. Please check that all references are relevant to the contents of the manuscript.

3.     Abbreviations that appear for the first time should be written with the full name.

4.     In section 2.1, “as shown in Fig. 2(a), the three different magnetic materials exhibited different metal-content changes.” It is not shown in the picture.

5.      In section 2.2, thermogravimetric analyzer should give enough details information.

I did notice a few errors in the English grammar. Most of them do not detract from the scientific understanding. But, for example, “In Figure 5(a), it shows the transition from ductile to brittle form the temperature of the first stage is the optimum magnetic transition temperature below the Curie point temperature, then increase the approximate crystallization temperature of the second stage and the complete crystallization temperature of the third stage and detect the amorphous by SEM” is hard to understand.

Author Response

Review Reply

2022/11/15

Title: Effect of Magnetic Flux Distribution and Noise of Local Annealing for Micro-Generator with Amorphous Shell

Manuscript ID: micromachines-1962722

Authors: Chang-Hung Hsu *Received: 25 September 2022 E-mails: chshiu@mail.aeust.edu.tw

Submitted to section: E:Engineering and Technology,

Dear Editor,

Thank you very much for your kind help. As this time, the reply letter and paper have revised as below.

Author: Chang-Hung Hsu

2022/11/15

Dear Reviewer

Thank you very much for valuable comments. The reply letter has revised as below.

The authors focus the paper on magnetic field properties of a generator with amorphous alloy shell. The topic is of interest. However, there exist some deficiencies in the papers. The main comments are listed as follow.

  1. The literature review is not enough, the conclusion of the review is quite rough and the authors' contributions can't be supported. Most of introduction underlines the torque, but it is unrelated to the main contents.

Reply: Thank you very much for valuable comments. This paper has been cited the relevant paper in this study, as shown in reference (Highlight with yellow color).

Therefore, reducing the zero-mode vibration of permanent-magnet synchronous motor (PMSM) is one of the most effective ways to meet the low vibration and noise requirements. Existing methods to reduce zero-mode vibration include electromagnetic optimization, mechanical design, and control strategies [9, 10]. Electromagnetic optimization mainly aims to reduce harmonics in the magnetic field by optimizing the shape of the permanent magnets (PMs) and stator slots [11]. In the literature, the correlation between electrical harmonics and torque generated in the air gap and the overall enclosure has been discussed [11-14]. It has been proposed that a brushless PMSM drive coupled with the DC generator and the three-phase inverter run synchronously; the harmonics of the high-voltage inverter can predict the torque vibration and the overall life of the housing. The output voltage and current of the three-phase inverter varies with high ripple harmonics, which will cause torque ripple in PMSM drives. Such ripples can have a considerable effect on the enclosure because of the abnormal torsional vibration and noise generated in the PMSM.

Further researches on these soft magnetic materials have considered various modifications to the production process, and additives or grain orientation modification have resulted in improved performances and reduced core power losses of up to 5% during operation at 50 Hz [15-16]. The cuboid soft magnetic bulk core made of amorphous FeSiB strips with a high saturation induction of Bs = 1.56 T, satisfactory mechanical strength, and relatively low power loss can successfully replace conventional Fe- Si steel production of harmonic filters. These filters are used in the grid during the transfer of electrical energy from the source to the object of application (i.e., the load). Their mission is to suppress harmonic currents flowing into the power system from the source by reducing harmful current distortion caused by nonlinear loads. So far, there are not many publications on the application of amorphous and nanocrystalline soft magnetic materials in harmonic filter choke coils. The main part of the filter is the choke coil assembly, which consists of a specific inductor L and capacitor C to ensure the desired suppression of higher harmonics. Literature [17, 18] studied the effect of material type (Fe-Si steel with 3 wt% Si; FeSiB) on the choke core power loss. Their results show that these losses are 7.7 W/kg in Fe-Si steel with 3 wt% Si core and mass of 18.5 kg at B = 0.8 T at a magnetizing field frequency of 350 Hz, while in  cores of the same mass are reduced to 0.96 W/kg. A general trend in magnetic material selection for output filter chokes of high frequency inverters (up to 150 kHz) powered by renewable energy sources is presented. The literature [19] have demonstrated the magnetic properties of bulk cores made from rapidly quenched tapes cast from a newly developed soft magnetic nanocrystalline alloy with FeCuMoSiB composition. This study compared the soft magnetic properties of nanocrystalline Fe-based bulk cores with amorphous FeSiB bulk cores and found that at B = 1.5 T and f = 600 Hz, the power loss of nanocrystalline and amorphous cores are 11 W/kg and 7 W/kg respectively. This proves that nanocrystalline materials are not significantly better than amorphous materials under such operating conditions, especially in terms of magnetic field frequencies. In another study [20], the results of applying nanocrystalline bulk cores produced from Fe-based ribbons in high frequency choke coils (10 kHz) are presented. Magnetic cores produced from nanocrystalline  ribbons for 25 kW DC-DC converters, rated to operate in the discontinuous conduction mode at 300 A peak current and switching frequencies of up to 20 kHz, have been described elsewhere [21, 22].

Reference:

  1. Ebrahimi, Y. Gao, A. Kameari, H. Dozono, and K. Muramatsu, “Coupled magneto-mechanical analysis considering permeability variation by stress due to both magnetostriction and electromagnetism,” IEEE Trans. Magnetics, vol. 45, no. 5, pp. 1621–1624, 2013.
  2. Y. Kim, G. H. Jang, and J. K. Nam, “Magnetically induced vibrations in an IPM motor due to distorted magnetic forces arising from flux weakening control,” IEEE Trans. Magnetics, vol. 49, no. 7, pp. 3929–3932, 2013.
  3. Suzana Uran,Božidar Bratina andRiko Šafarič, A microfluidic rotational motor driven by circular vibrations, Micromachines, vol. 10, no. 12, pp. 809, 2019.
  4. Wenxing Chen, Shuyang Dai and Baojuan Zheng, “A dynamic thermal-mechanical coupling numerical model to solve the deformation and thermal diffusion of plates,” Micromachines, vol. 13, no. 4, pp. 753-1-36. 2022.
  5. Oka, T. Ogasawara, N. Kawano, and M. Enokizono, “Estimation of suppressed iron loss by stress-relief annealing in an actual induction motor stator core by using the excitation inner core method,” IEEE Trans. Magnetics, vol. 50, no. 11, pp. 1–4, Nov. 2014.
  6. Chai, P. Liang, Y. Pei, and S. Cheng, “Analytical method for iron losses reduction in interior permanent magnet synchronous motor,” IEEE Trans. Magnetics, vol. 51, no. 11, 2015.
  7. Gao, G. Xu, X. Guo, G. Li, Y. Wang, Primary recrystallization characteristics and magnetic properties improvement of high permeability grain-oriented silicon steel by trace Cr addition, JMMM, vol. 507, no. 1, pp.166849, 2020.
  8. Zhang, H. Gu, S. Yang, A. Huang, Improved magnetic properties of grainoriented silicon steel by in-situformation of potassium zirconium phosphate in insulating coating, JMMM, vol. 506, no. 15, p.166802, 2020.
  9. Cheng, G. Ma, X. Chen, F. Yang, L. Meng, Y. Yang, G. Li, H. Dong, Evolutions of microstructure and magnetic properties of heatproof domain-refined silicon steel during annealing and its application, JMMM, vol. 514, no. 15, p. 167264, 2020.
  10. Wang, S. Zhang, Z. Sun, L. Sun, Structure Design and Properties of Amorphous Filter Reactors, Materials and Manufacturing Processes, vol. 27, no. 11, p. 663138, 2012.
  11. Yao, M. Inoue, K. Tsukada, F. Fujisaki, Soft Magnetic Characteristics of Laminated Magnetic Block Cores Assembled With a High Bs Nanocrystalline Alloy, AIP Adv. Vol. 8, p. 056640, 2018.
  12. Soi´nski, J. Leszczy´nski, C. ´Swieboda, M. Kwiecie´n, Nanocrystalline Block Cores for High- Freguency Chokes, IEEE Trans. Magnetics, vol. 50, no. (11. pp. 1–4, 2020.
  13. Long, M. McHenry, D.P. Urciuoli, V. Keylin, J. Huth, T.E. Salen, Nanocrystalline Material Development for High-Power Inductors, J. Appl. Phys. Vol. 103, 07E705, 2008.
  14. Bertotti, General Properties of Power Losses in Soft Ferromagnetic Materials, IEEE Trans. Magnetics, vol. 24, pp. 621–630, 1988.

5.        Conclusion

This study considered a micro-alternator operated with different housing structures: an uncoated shell structure and a shell-coated generator made of an iron-based amorphous-alloy soft magnetic material. The electromagnetic power and noise characteristics of generators with different shell structures were measured and analyzed. The amorphous-alloy material SA1 was used for the shell coating. The experimental shell was between the above magnetic phase-transition characteristics, included as paramagnetic and ferromagnetic phases. The magnetic fluxes of the magnetic phase transitions of the amorphous shell were measured by a flux meter. The magnetic flux in the center of the amorphous shell was 1.2–2.4 mT, whereas the magnetic flux distributed around the shell was 0.6–1.0 mT. The flame-annealing temperature at the local center point of the amorphous casing reached the Curie temperature, and the noise of the casing could be reduced by at least 15 dB compared with that of the non-casing generator. Additionally, for a higher Curie (Tc) shell compared to the annealing shell, the reduction of the noise was ~30 dB. However, the overall performance of the generator with amorphous shell, in terms of harmonics and power, was not significantly improved.

  1. In page 2, line 12, “In the literature, an LSM…..” , which literature? It lacks the necessary references. Please check that all references are relevant to the contents of the manuscript.

Reply: Thank you very much for valuable comments. This paper has been revised as item 1.

  1. Abbreviations that appear for the first time should be written with the full name.

Reply: Thank you very much for valuable comments. The paper has been revised.

  1. In section 2.1, “as shown in Fig. 2(a), the three different magnetic materials exhibited different metal-content changes.” It is not shown in the picture.

Reply: Thank you very much for valuable comments. The paper has been revised as below. (Highlight with water blue color)

2.  Measurement and Analysis of Ferromagnetic and Paramagnetic Properties of Amorphous Alloy Shells

This study aimed to measure the magnetic properties of FeSiB ribbons at different annealing temperatures, in order to further understand the correlation between the two through microstructure analysis and observation. Firstly, the thermal properties of the ribbons were analyzed by differential scanning calorimetry (DSC), then the phase composition of the annealed ribbons was identified by X-ray diffraction (XRD). Scanning electron microscopy (SEM), which uses a highly focused electron beam to scan the surface of the sample, was used to observe the microstructure of the thin strip section. The brittleness of the ribbon was determined by the bending test to determine the brittleness temperature of the ribbon after annealing, and the vibrating-sample magnetometer (VSM) and the magnetostriction meter were then used to measure the hysteresis curve of the ribbon and the amorphous-material length changes.

  • High Temperature Magnetic Transitions by DSC

By using magnetic meter, the SA1 metal-content () is measured by inductively coupled plasma mass spectrometry (ICP-MS). Thermogravimetric analysis (TGA) is a method of changing the physical and chemical properties of a substance with an increase in temperature (isothermal heating rate) or time (isothermal and/or mass conservation loss). A differential scanning calorimeter (MDSC 2920, TA Instruments) was used to analyze 11-mg samples of SA1 amorphous alloy between 25 and 800 °C at a temperature increase rate of 5 °C/min. The DSC measurement results in this experiment are shown in Fig. 2(a). This iron-based amorphous alloy has a crystallization exothermic peak, and when the heating rate increases, the crystallization peak shifts to a higher temperature. This shows that the amorphous-alloy material has a direct correlation between composition and crystallization behavior. As shown in Figure 2(a), the exothermic peak is strongly influenced by the Fe and B contents; when these are higher, the crystallization peak also rises. The measurements were acquired at low temperature. The crystallization peak temperature of Fe-based FeSiB amorphous ribbon was 482.6 °C, and the crystallization activation energy was measured to be 19.5 J/g.

(a)                              (b)

Figure 2. Magnetic properties of amorphous alloy: (a) exothermic heat flow by scanning calorimetry and (b) thermogravimetric analysis.

  1. In section 2.2, thermogravimetric analyzer should give enough details information.

Reply: Thank you very much for valuable comments. The paper has been revised as below. (Highlight with green color)

  • Material Weight Change and Curie Temperature

An amorphous magnetic material must measure the Curie temperature () process of the phase change that reaches a specific temperature. Thermogravimetric analysis (TGA) was performed using a thermogravimetric analyzer to observe the weight change in the physical and chemical properties of the materials as the temperature (equal heating rate) or time (isothermal and mass conservation loss) increased, as shown in Figure 2(b). The TGA test result is of SA1 at 385.6 °C.  This result was the phase-transition process of the magnetic material and is more stable for annealing temperature processing. It was indicated that ~360 °C is suitable for the annealing owing to the  effect.

  1. I did notice a few errors in the English grammar. Most of them do not detract from the scientific understanding. But, for example, “In Figure 5(a), it shows the transition from ductile to brittle form the temperature of the first stage is the optimum magnetic transition temperature below the Curie point temperature, then increase the approximate crystallization temperature of the second stage and the complete crystallization temperature of the third stage and detect the amorphous by SEM” is hard to understand.

Reply: Thank you very much for valuable comments. The paper has been revised as below. (Highlight with yellow color)

  • Relationship between Phase Crystallization Peaks and Hysteresis Loops

Figure 4 shows the results of XRD analysis of the phase structure of the ribbon before and after annealing. The figure indicates that the annealing temperature below 400° is still the same as the diffraction result of the unannealed thin strip, showing a broad peak of 2θ within 40°–50°, which is evidently amorphous. Until the 400° point, a small diffraction peak of α-Fe (110) appears at 44.6°. After the annealing process, the crystalline state of the material was checked, and the properties of the material were observed using an X-ray diffractometer (D8 DISCOVER SSS Multi-Function High Power X-ray Diffractometer). The sharp peak in Figure 4(a) indicates that α-Fe especially exhibits strong crystallization between annealing temperatures from 360 °C to 450 °C. At a higher temperature of 450 °C, not only the crystalline signal of α-Fe becomes stronger, but also the diffraction peaks of crystalline phases such as  and  begin to appear. This results in terms of the reduction of the saturation magnetic flux density and magnetic permeability of the magnetic material at 360 °C are shown. However, unexpectedly, the vibration and noise between 390 °C and 450 °C were also reduced to a low level, owing to the deterioration of the magnetic properties. The rate of change in lagging expansion and contraction was evidently slower. Furthermore, the annealing temperature of the amorphous-alloy material is between 360 °C and 450 °C, and the hysteresis loop exhibits different changes. This XRD diffraction analysis result corresponds to the curve measured by the previous DSC. It can be inferred that the single peak in the DSC curve represents the superposition of two crystallization reactions. The crystallization reaction temperature ranges of Fe and  are remarkably close, and there is a possibility of overlap during the slow heating process that tends to be stable. However, the difference between the two crystallization reactions would be increasingly prominent under the condition of faster heating rate. This may explain the correlation between the two sets of data from XRD and DSC.

By measuring the VSM hysteresis curves at different annealing temperatures, different parameters such as saturation magnetization () and coercivity () can be obtained, as discussed in subsequent sections. The BH measurement results at different annealing temperatures are plotted in Figure 4(b). The amorphous ribbon has higher Bs and lower  when the annealing temperature at 360 °C is lower than . On the contrary, if the annealing temperature is higher than Tc, the crystalline state would gradually appear. At this time, Bs decreases and  increases. Increasing the annealing temperature causes the area of the B-H ring to become larger. The difference between the two can explain the property change caused by the structural change of the thin strip after heat treatment.

(a)                                    (b)

Figure 4. X-ray diffraction measurement results: (a) Sample SA1 crystallization peak at different temperatures and (b) Sample SA1 to crystallization peak and α-Fe at different temperatures.

  • After Annealing Material Brittle and Ductile Microstructure

The specimen used in the experiment is a thin FeSiB amorphous ribbon with a thickness of about 25 μm. The raw material strips were annealed at 350, 390, and 450 °C for 120 min under nitrogen protection. The brittleness of the strip after the annealing treatment was determined using bending tests. The strips were cut into approximately 1 cm × 6 cm at room temperature, bent into a U-shape, and placed between two parallel plates. After the parallel plates were slowly pressed together, a reference for the cross-section of the thin strip of material could be observed after it becomes brittle. After pressing slowly, the distance d is measured between the two plates when the thin strip breaks. The reference strain  when the thin strip is broken can be estimated and calculated by the following simple formula as reference data for assessing the brittleness of the thin strip. The same test piece is tested five times to find the average rupture strain value, as follows:

                                                                            (1)

where  is the thickness of the thin strip, which is known to be about 25 μm. It is defined that when  < 1.0, the thin strip begins to appear embrittlement, which can be used to determine the brittle transition temperature of the thin strip after annealing.

The microstructure and material properties of the materials were obtained using SEM (JEOL, JSM-6500F). Different annealing temperatures were controlled between 350 and 450 °C for the ductile and brittle phase transitions, as shown in Figure 5(a). As shown in Fig. 5(b), the SEM reveals the transition from ductile to brittle, where the temperature of the first stage is the optimal magnetic transition temperature below the Curie point temperature in the body-centered cubic (BCC) α-Fe. The phase transformation of Fe, bending the material in half into a U-shape is about the cross-section formed by folding in half 0.2mm.  Then, as shown in Fig. 5(c), the temperature of the second stage is increased slightly beyond the Curie point, the crystal material is gradually converted from α-Fe of BCC to FCC, and the material is bent into a U-shaped cross-section formed by folding in half about 0.5 mm situation. The third stage is to increase the annealing temperature of the material to approximately the crystallization temperature, resulting in the complete FCC crystallization condition of the material. As shown in Figure 5(d), the cross-section was formed by bending the material in a U-shape by about 10mm. The annealed iron ribbon gradually loses its magnetic properties and is almost crystalline, and the physical properties of hysteresis are also gradually lost. The measurement of the brittleness of the thin strip is carried out via the bending test.

At room temperature, the thin strip is cut into strips of about 1cm×5cm, bent into a U-shape and placed between two parallel plates. When the rupture strain of the ribbon is equal to 1, the ribbon is still in a ductile state, but when the rupture strain is less than 1, the ribbon becomes brittle. It can be seen from the figure that the temperature at which the thin strip starts to become brittle is between 360 °C and 390 °C. With the increase of the annealing temperature, the fracture strain decreases rapidly and the deviation value is large at this time. When the annealing temperature is 450 °C, which is the crystallization temperature, the rupture occurs. The strain approaches zero and the deviation value approaches zero. The above change trend is also in line with the change trend of the hermos-electromagnetic properties measured previously, that is, the effect of annealing on the mechanical properties can be divided into two stages: the short-range ordered structure produced by low-temperature annealing causes the ribbon to become brittle. However, it is only partially brittle, which may be because of the difference between the stress and strain of the short-range ordered structure and the original disordered amorphous structure.; when the difference is even greater, its interface becomes the fracture-inducing region of stress concentration.

(a)                           (b)

(c)                           (d)

Figure 5. Measurement of the microstructure of material sample 3: (a) Measurement of ductility and brittleness temperature range after annealing; SEM images at (b) 350 °C, (c) 390 °C, and (d) 450 °C.

Reviewer 2 Report

The paper Can be accepted now 

Author Response

(The authors gave the same response as above.)

Reviewer 3 Report

Thank you for the manuscript improvements. The paper contribution is well represented in the revised version.

Author Response

(The authors gave the same response as above.)
